# Quality Assessment of Untargeted Analytical Data in a Large-Scale Metabolomic Study

**DOI:** 10.3390/jcm10091826

**Published:** 2021-04-22

**Authors:** Rintaro Saito, Masahiro Sugimoto, Akiyoshi Hirayama, Tomoyoshi Soga, Masaru Tomita, Toru Takebayashi

**Affiliations:** 1Institute for Advanced Biosciences, Keio University, Tsuruoka, Yamagata 997-0052, Japan; msugi@sfc.keio.ac.jp (M.S.); hirayama@ttck.keio.ac.jp (A.H.); soga@sfc.keio.ac.jp (T.S.); mt@sfc.keio.ac.jp (M.T.); ttakebayashi@keio.jp (T.T.); 2Health Promotion and Preemptive Medicine, Research and Development Center for Minimally Invasive Therapies, Tokyo Medical University, Tokyo 160-8402, Japan; 3Faculty of Environment and Information Studies, Keio University, Fujisawa, Kanagawa 252-0882, Japan; 4Department of Preventive Medicine and Public Health, Keio University School of Medicine, Tokyo 160-8582, Japan

**Keywords:** cohort study, metabolomics, capillary electrophoresis-mass spectrometry

## Abstract

Large-scale metabolomic studies have become common, and the reliability of the peak data produced by the various instruments is an important issue. However, less attention has been paid to the large number of uncharacterized peaks in untargeted metabolomics data. In this study, we tested various criteria to assess the reliability of 276 and 202 uncharacterized peaks that were detected in a gathered set of 30 plasma and urine quality control samples, respectively, using capillary electrophoresis-time-of-flight mass spectrometry (CE-TOFMS). The linear relationship between the amounts of pooled samples and the corresponding peak areas was one of the criteria used to select reliable peaks. We used samples from approximately 3000 participants in the Tsuruoka Metabolome Cohort Study to investigate patterns of the areas of these uncharacterized peaks among the samples and clustered the peaks by combining the patterns and differences in the migration times. Our assessment pipeline removed substantial numbers of unreliable or redundant peaks and detected 35 and 74 reliable uncharacterized peaks in plasma and urine, respectively, some of which may correspond to metabolites involved in important physiological processes such as disease progression. We propose that our assessment pipeline can be used to help establish large-scale untargeted clinical metabolomic studies.

## 1. Introduction

Large-scale metabolomic studies using human samples, including cohort studies, have been conducted to discover metabolites associated with specific biochemical or physiological conditions. Identification and quantification of a large number of metabolites are the most important procedures, and quality control of the observed data is also an ineluctable process for obtaining reliable and reproducible data. Among the various metabolite profiling technologies, nuclear magnetic resonance and mass spectrometry (MS) have been commonly used in various cohort studies [1].

Two types of analytical approaches have been used for profiling metabolites. Targeted analysis is an established approach that has been used to analyze pre-defined metabolite sets. Usually, a standard compound for each metabolite is used to identify the corresponding peak, and compound-based quality assessment methods have been developed [2,3]. Conversely, untargeted metabolomics approaches aim to analyze all observed signals measured by an instrument [4]. However, this approach has two major challenges: (1) identifying the metabolite for each peak (uncharacterized peaks), and (2) quality control of the quantification accuracy of all observed data. In this study, we focused on the quality control issue, which has been less of a focus in previous studies. We conducted a reliability assessment of uncharacterized peaks from untargeted metabolomics data to determine whether they can be used per se as surrogates for the biochemical or physiological conditions of a given sample without the need to identify the corresponding metabolites.

The discovery of metabolites associated with important physiological processes is greatly influenced by the numbers of samples and types of metabolites. With untargeted metabolomics, huge numbers of peaks can be produced. Although some of the uncharacterized peaks can be correlated with cellular conditions or physiology conditions, including diseases [5,6,7], most studies have focused on analyzing characterized peaks, and uncharacterized peaks have been ignored [8]. This is because the biological interpretation of characterized peaks is much easier, and usually they have higher priority when the number of metabolites that can be investigated is limited.

To analyze uncharacterized peaks efficiently, the difficulties of quality assurance and quality control of untargeted metabolomics data must be addressed, and the quantification of as many metabolites as possible must be attempted. The large numbers of peaks detected from untargeted metabolomics contain a lot of noise [1,9,10]. Additionally, because the instruments are often sensitive to various factors, metabolite levels inferred from the peaks in chromatograms or electropherograms can vary depending on the laboratory, machine, or even the date when the measurement was made [11,12]. Therefore, how to assure the quality of as many peaks as possible while monitoring their patterns over sets of measurements (i.e., batches) is an important issue [13].

Automation of peak processing has been used to measure as many peaks as possible, and various processing tools have been developed to automatically extract peaks from raw data [14,15]. However, it has been argued that visual inspection and manual curation of the peaks may be necessary to ensure their quality [16,17].

Pooled quality control (QC) samples have been prepared and measured regularly within and across batches to ensure that measurements based on detected patterns of peaks were stable [18]. Peaks in QC samples must be representative of the peaks in actual samples to check that each of the peaks that usually exists in the actual samples is stably detected in the QC samples. Therefore, the number of peaks that can be analyzed reliably is restricted by the number of metabolites in QC samples and possibly by the peak extraction method, that is, whether it is automated or manually curated. For targeted metabolomics, the number of metabolites also is restricted by the available standard compounds. To determine whether the number of target peaks, including uncharacterized ones, can be expanded to increase peak coverage with reasonable reliability and without a curation cost, we assessed the reliability of uncharacterized peaks from capillary electrophoresis time-of-flight-mass MS (CE-TOFMS) that were extracted automatically from the samples.

In a preliminary study, we customized our pipeline for peak assessment to analyze metabolome data from approximately 3000 subjects who participated in the Tsuruoka Metabolome Cohort Study (TMCS). The study was initiated in April 2012 with approximately 10,000 participants from Tsuruoka, Yamagata Prefecture, Japan, with the aim of conducting a longitudinal investigation of the metabolome profiles of the participants. Plasma and urine samples from the participants and the standard solution were measured by CE-TOFMS. The obtained peaks were compared manually with those in the standard solution to identify and quantify each metabolite in each sample. The reliability of the characterized peaks from the plasma samples was assessed using the QC and actual samples [2], and several correlation analyses between metabolites and clinical values were conducted [19]. However, because the numbers of available standard compounds and analysts to curating peaks were limited, only approximately 115 metabolites have been analyzed so far. In the present study, we developed a pipeline to assess reliabilities of automatically extracted peaks from given samples (Figure 1).

Assessment of variations between curated and characterized peaks over different batches has been reported previously [2]. In the present study, we evaluated the reliability of the uncharacterized peaks using multiple approaches including rates of peak detection in 30 randomly chosen QC samples, accuracy of quantification according to independently pooled/diluted samples, and whether the uncharacterized peaks in the 30 randomly chosen QC samples reflect those in the actual samples regarding the signal-to-noise (S/N) ratio. We also investigated patterns in peak areas among the samples to find clusters of peaks with similar patterns and therefore likely corresponding to single metabolites. Automatic peak extraction was one of the important procedures for our pipeline (Figure 1c), and we used our proprietary software MasterHands [6,20] for this purpose. The computational accuracy of the automatic peak extraction was estimated by comparing the area of characterized peaks that were extracted and manually curated with those that were extracted automatically.

Although our assessment pipeline estimated that there were a substantial number of unreliable uncharacterized peaks, it identified 35 and 74 reliable peaks that were extracted automatically from plasma and urine samples, respectively, thus expanding the current number of characterized peaks (approximately 115) that have been investigated in the TMCS so far. The increased number of reliable peaks will increase the chances of discovering other important peaks.

## 2. Materials and Methods

### 2.1. Data Collection and Computational Analyses Pipeline

An overview of the data collection and computational analyses pipeline is shown in Figure 1a. Plasma and urine samples were collected from the participants in the TMCS. Participants were asked for an overnight fasting (12 h) in the night immediately before the sample collection to avoid variation due to fasting state and circadian rhythm. Plasma samples were collected in the morning with ethylenediaminetetraacetic acid-2Na (EDTA-2Na) as an anticoagulant and kept at 4 °C immediately after collection. The samples were centrifuged for 15 min (1500× *g* at 4 °C) within 3 h of collection, divided into aliquots, and preserved at 4 °C until extraction of metabolites. Metabolite extraction from plasma was finished within 6 h after collection to reduce any metabolic reactions in plasma, and then the extract was stored at −80 °C.

To extract metabolites from plasma samples, 50 μL aliquots were put into 450 μL of methanol that contained internal standards (20 μmol/L each of methionine sulfone and camphor 10-sulfonic acid). The internal standards were used to normalize the extraction efficiency of the metabolites during sample preparation and to calculate the concentration for each metabolite. The solutions were mixed well, and 500 μL of chloroform and 200 μL of Milli-Q water were added, followed by centrifugation at 4600× *g* for 5 min at 4 °C. Then, 150 μL of the aqueous layer was transferred to a 5-kDa cutoff centrifugal filter tube (Human Metabolome Technologies, Yamagata, Japan) to remove proteins. The filtrate was concentrated centrifugally at 40 °C and reconstituted with 50 μL of Milli-Q water that contained reference compounds (200 μmol/L each of 3-aminopyrrolidine and trimesic acid) immediately before CE-TOFMS analysis. These reference compounds were added to eliminate the variation in migration time of individual peaks in electropherograms among multiple datasets.

Urine samples also were collected in the morning between 8:30 am and 10:30 am after overnight fasting. They were frozen at −80 °C after collection and thawed before sample preparation. The samples were initially vortexed for 30 s, followed by centrifugation at 2300× *g* for 5 min at 4 °C. The supernatants were transferred and diluted according to the creatinine concentration with Milli-Q water and Milli-Q water containing internal standards (2 mM each of methionine sulfone, camphor-10-sulfonic acid, 3-aminopyrrolidine, and trimesic acid). The diluted samples were filtered through a 5-kDa cutoff centrifugal filter tube (Human Metabolome Technologies). We found that a creatinine concentration of <10 mg/dL did not cause ion saturation in the mass spectrometer. Hence, we set this as the upper limit for diluting the urine. Additional details of sample processing and metabolome analyses by CE-TOFMS have been described previously [5,6,21,22]. Target peaks corresponding to approximately 115 metabolites were manually curated by two analysts as part of the general routine in TMCS (referred to as “TMCS core” in Figure 1a) that is described in [19]. For the present study, we organized the baseline metabolome profiles of approximately 3000 of the participants and used them in our pipeline for automatic extraction and assessment of uncharacterized peaks. Information related to the metabolome analyses, including information about samples, batches, and peaks, is stored in our in-house database (SQLite3, available on a collaboration basis). The sample information includes specimen type (plasma or urine). The batch information includes measurement dates, analysts, modes of CE (cation or anion), and ID of the instrument used for the CE-TOFMS. The stored peak features includes *m/z*, migration time, intensity, area, area relative to the area of internal standards (RelArea), and S/N ratios. Manually curated and automatically extracted peaks with their annotations were imported into the database. A simple web-based graphical user interface was implemented using the Django interface to browse data in the SQLite3 database. Scripts for data processing and statistical analyses were written in Python or R.

### 2.2. Determining the Target Peaks

To define target peaks with a focus on uncharacterized peaks, we collected CE-TOFMS data from 30 randomly chosen QC samples (referred to as gathered QC samples) all from different batches (Figure 1b). The gathered QC samples were prepared by pooling the plasma or urine samples from 10–20 participants and dividing them into aliquots; therefore, the metabolic profiles of the QC samples could be identical. The QC samples were intended to be used to monitor the stability and variances of the measurements. The peaks with intensities above the S/N threshold (S/N >5) in the gathered QC samples were detected using our proprietary software MasterHands [6,20]) and matched (aligned) among the samples to form peak groups (Figure 1b). This procedure was done manually for cation and anion CE modes in the plasma and urine samples. After curation, the detected peak groups were used as the initial list of target peaks in this study. We estimated the reproducibility of each peak as reflected by the rate of the peak detections in each peak group. Then, we compared the average *m/z* and migration time of each peak group with the list of migration times and *m/z* values of the metabolites that were determined using the standard compounds to find peak groups that match the peaks of known metabolites. The matched peaks were considered “known”, and the peaks without matches were considered “uncharacterized peaks”; the migration times and *m/z* values were included in the peak annotations. Using the MasterHands software, peaks detected with their isotope or adduct ion peaks were considered “parent peak”, and these isotopes and adduct ion peaks were removed from further analyses.

### 2.3. Automatic Peak Extraction

After the target peaks had been determined, the peaks in each sample were extracted automatically without manual curation to obtain as many peaks as possible while saving time. This step is important for scalability, especially for projects that generate a large number of samples and metabolites. The automatic peak extraction procedure follows the standard procedure of raw data processing in metabolomics [23] using the MasterHands software, which also has been used in other studies [24,25]. A detailed description of the MasterHands algorithm was provided in [6]. Briefly, electropherograms are extracted from measurements based on *m/z* of target peaks, and peaks with S/N ratios above the threshold are chosen from each electropherogram. Then the electropherograms from the samples and standard solution in each batch are aligned to those of the reference sample using the normalization function for CE-migration [26] after optimization [27,28,29], and the detected peaks are matched across the samples to form peak groups. For each group, averaged migration times and *m/z* values are calculated and matched to those of target peaks so that the groups that correspond to target peaks can be identified. For semi-quantification, the peak area relative to the peak areas of internal standards (relative peak area, RelArea) was calculated for each peak. RelArea was used to reflect the metabolite level in the samples.

To assess the quantitative accuracy of this automatic procedure, we used the RelAreas of the known metabolites that were manually curated and had accumulated in the TMCS. We compared the RelAreas of the known metabolites calculated using this automatic procedure with the RelAreas of metabolites that had been manually curated.

### 2.4. Dilution of Independently Prepared and Pooled Samples

If a peak originated from a sample, not from noise, then the area of that peak should inversely decrease with dilution of the sample (i.e., the relationship between the amount of sample and the peak area should be linear and positive). To confirm this relationship, the eight prepared plasma samples were pooled and diluted as follows: ×10, ×20, ×39, ×83, and ×167; the corresponding amounts of samples were 50, 25, 13, 6, and 3 μL, respectively. Dilution of samples was conducted before any other sample preprocessing so that only metabolites in the samples, not in reagents (i.e., buffers), were diluted. This procedure is summarized in Figure 1a (upper-right) and Appendix A. Similarly, urine samples from 15 subjects were pooled (Appendix A) and diluted as follows: ×5, ×10, ×20, ×40, and ×100; the corresponding amounts of the samples were 40, 20, 10, 5, and 2 μL, respectively. Each diluted sample was measured in triplicate. Pearson’s correlation coefficients between the amounts of samples and peak areas were then calculated to assess the authenticity of each peak. Pearson’s correlation coefficient also can be an indicator of how well the peak area reflects the metabolite level.

### 2.5. Clustering Peak Fragments

A single metabolite may be fragmented during ionization, and peaks that originated from a single metabolite may have similar migration times. Because the areas of these peaks reflect the amount of the single metabolite in a sample, the areas of these peaks are expected to be positively correlated over the samples. Therefore, we clustered peaks that had similar migration times and relative areas that correlated over the samples. Complete linkage hierarchical clustering with the distance metric max (1—correlation coefficient, the difference in average migration time × weight) was calculated to generate clusters based on the threshold of linkage score.

## 3. Results

### 3.1. Defining Target Peaks Using the Gathered QC Samples

In the general routine of the TMCS (TMCS core in Figure 1a), one batch comprised approximately 100 samples, and one QC sample was placed in every 10 actual samples for the measurements. In the present study, we used QC samples that were collected from 30 randomly selected batches. Peaks with S/N ratios >5 in the samples were extracted and their electropherograms were aligned (Figure 1b). We detected 340 and 282 peak groups in plasma and urine samples, respectively. Then, we compared the average migration times and *m/z* values of these peak groups with the values of known metabolites to find peak groups that matched the known metabolites. A total of 64 and 80 peak groups in the plasma and urine samples, respectively, matched those of known metabolites, and 38 and 51 of them were assigned unambiguously to single metabolites and considered known peaks. The remaining 276 and 202 peak groups in the plasma and urine samples, respectively, were considered uncharacterized peaks. We checked the spectrum of each uncharacterized peak to determine whether neighboring peaks that could represent isotope ions or adduct ions of a metabolite could be found. Peaks that matched such ions were considered parent peaks, and the remaining peaks were classified as singletons.

Because the QC samples were identical, the expected detection rate of each peak among the gathered QC samples was 100%. However, the actual detection rate varied between 0.4 and 1.0 (Figure 2) because of the inefficiency of alignment caused by large variance in migration times or mis-extraction of peaks because of noise in the electropherogram. Still, about half of the peaks had detection rates of >90%. The uncharacterized peak groups tended to have lower peak detection rates, which reflects the observation that uncharacterized peaks were less reproducible than known peaks. Furthermore, the peak groups that had the higher detection rates tended to have higher S/N ratios (Appendix A).

Among the known peaks, parent peaks tended to have higher peak detection rates, whereas this tendency was weaker for uncharacterized peaks. Therefore, detection of potential isotope or adduct ions may not be effective for finding reproducible uncharacterized peaks.

### 3.2. Dilution of Independently Prepared Pooled Samples

We evaluated the reproducibility of each uncharacterized peak in the gathered QC samples using independently prepared pooled samples. Among the 276 and 202 uncharacterized peaks, 181 and 180 of them were found in the independently prepared pooled plasma and urine samples, respectively, with S/N >5. There was a linear relationship between the average S/N ratios in the gathered QC and independently pooled samples (Appendix A), which implies that the signal strengths of the peaks were similar in these two independent sets of pooled samples.

Next, we assessed whether the area of the peaks was quantitative, that is, whether there was a linear relationship between the area of the peaks and the number of corresponding metabolites in a sample, which implies that the areas of the peaks that originated from metabolites in the sample were proportional to the metabolite concentrations. To confirm this linear relationship, we diluted the independently prepared pooled samples and calculated Pearson’s correlation coefficients between the amounts of samples and peak areas to the area of internal standards for each peak (relative peak area, RelArea). The correlation coefficients for known peaks were concentrated around 0.8 for plasma, and 0.6 and 1.0 for urine (Figure 3a). The correlations coefficients for uncharacterized peaks had a broader distribution, although for most of the peaks with positive correlations, the coefficients were >0.3. In particular, the areas of 115 out of the 181 peaks in the independently prepared pooled plasma samples and 153 out of the 180 in the independently prepared pooled urine samples had correlation coefficients >0.3 with the amounts of plasma and urine samples, respectively.

The results for some representative examples are shown in Figure 3b. One of the known metabolites, histidine, was detected in the three types of standard solutions as expected, and the relative area increased as the amount of sample increased (Pearson’s correlation coefficient = 0.758). For the uncharacterized peak mz106.95mt5.8, its relative peak areas in the standard solutions were low, which suggests that the standard solutions were unlikely to contain mz106.95mt5.8, which implies that this peak was unlikely to be one of the known metabolites. A positive linear relationship was detected between the amounts of sample and relative peak areas (Pearson’s correlation coefficient = 0.662), and thus mz106.95mt5.8 was quantitative. Conversely, no positive linear relationship was detected between the amounts of sample and relative peak areas for mz115.0682mt9.72, and thus mz115.0682mt9.72 was considered to be unreliable.

### 3.3. Automatic Peak Extraction from Actual Samples

Using our initial list of target peaks, we automatically extracted peaks from the approximately 3000 samples in the TMCS. The list included known metabolites with S/N ratios above the threshold. To estimate the accuracy of the automatic extraction, relative peak areas of known metabolites that were extracted automatically were compared with those that were manually curated. We found that 20 out of 27 metabolites and 35 out of 37 metabolites had Spearman’s correlation coefficients >0.8 in the plasma and urine samples, respectively (Table 1), which indicated the accuracy of our automatic procedure was comparable to that of manual curation. The results for one of the known metabolites, serine, is shown in Figure 4a. For some metabolites, no remarkable positive correlations were found (e.g., phenylalanylphenylalanine, Phe–Phe, in plasma), indicating that it was very difficult to correctly identify peaks of these metabolites, mostly because of low peak height, high surrounding noise, or variations in migration times. Other metabolites, such as malate or 5-oxoproline in plasma, had high Spearman’s correlation coefficients but low Pearson’s correlation coefficients mainly because of outliers in the data. In such cases, there was great discrepancy between the peak chosen automatically and the peak chosen manually from a given electropherogram. Thus, the risk of picking a wrong peak when multiple peaks are present in a single electropherogram should be considered.

We checked whether the features of uncharacterized peaks in the gathered QC samples were similar to those in the actual samples. In particular, we expected that the intensities of peaks in the gathered QC samples would, on average, reflect those in the actual samples. If they did not, then the peak profiles in the QC samples may be very different from those in the actual samples. We found a strong positive correlation between average S/N ratios of the uncharacterized peaks in the gathered QC samples and those in the actual samples (Figure 4b). This result confirmed that the S/N ratios of the peaks in the QC samples were representative of those in the actual samples, which implies that peaks found in the QC samples also were likely to be found in the actual samples.

To check whether the areas of uncharacterized peaks in QC samples measured in batches with the actual samples were fairly constant, we calculated within-batch and inter-batch coefficients of variations (CVs) of the peaks (Figure 4c). For plasma, 28 out of 38 known peaks (73.7%) had both within-batch and inter-batch CVs <0.2, whereas only 43 out of 276 uncharacterized peaks (15.6%) had both CVs <0.2. For urine, 31 out of 51 known peaks (60.8%) had both within-batch and inter-batch CVs <0.2, whereas 53 out of 202 uncharacterized peaks (26.2%) had both CVs <0.2. Thus, the relative areas of known peaks in the QC samples were much more stable than those of the uncharacterized peaks. However, this result is expected because in the general TMCS routine, the results of the metabolome measurement for a batch is discarded and the samples in the batch are re-measured if the area variations of the specific set of known peaks are outside the mean ± two standard deviations.

### 3.4. Clustering Peaks Based on Inferred Origin Metabolites

A single metabolite can generate multiple fragments during the ionization process, which appear as fragment peaks with similar migration times. Their peak areas also are likely to be correlated among the samples because they originated from the same metabolite. To capture the characteristics of this relationship, we calculated correlations of relative area (RelArea) of peak pairs over the samples and differences in their migration times (dMTs) for all peak pairs. Figure 5a shows the relationship between the correlation coefficients and dMTs. For plasma, the distribution of dMTs was broad for peak pairs with correlation coefficients of their areas among the samples of <0.5. However, for peak pairs with correlation coefficients ≥0.6, most of the dMTs suddenly dropped to <0.75. Thus, peak pairs with correlation coefficients ≥0.6 were likely to have close migration times, presumably reflecting that each peak pair originated from the same metabolite, which fragmented at a specific migration time before their *m/z* was analyzed by mass spectrometry. For urine, most of the dMTs dropped to <0.2 at correlation coefficients >0.95. The results for representative uncharacterized peak pairs with close migration times and high correlations among samples are shown in Figure 5b.

We clustered the peaks on the basis of these results so that peaks in the same cluster had similar migration times (dMT < 0.75 for plasma and <0.2 for urine) and their peak areas were correlated (correlation coefficient > 0.6 for plasma and >0.95 for urine). Among the 115 and 153 peaks that were initially uncharacterized and whose areas were positively correlated with the amounts of samples, 13 and 8 of them clustered with known peaks from plasma and urine, respectively. Among the rest of the peaks, 53 and 114 formed clusters with other peaks from plasma and urine, respectively. Overall, approximately 67% and 34% of the uncharacterized peaks clustered with other peaks in plasma and urine, respectively.

### 3.5. Peak Reliabilities and Their Relevant Factors

Finally, we conducted comprehensive analyses of possible factors that may influence the peak reliabilities. Reliability was evaluated based on two aspects: one was the linearity of the relationship between the amounts of samples (determined by dilution rates) and the corresponding relative peak areas (RelAreas) as measured by Pearson’s correlation coefficient; and the other was an assessment of how well the variations of relative peak areas in the QC samples were restrained both within each batch and across the batches (inter-batch) when the QC samples were measured with actual samples in the batches. The restrained coefficient of variation was defined as −1 × CV, where a high CV indicates the variation is restrained at a high level. Assuming that these factors (linearity of the relationship between the amounts of samples and the corresponding relative peak areas, and variations of relative peak areas in the QC samples restrained both within-batch and inter-batch) represent peak reliabilities, we calculated the influence of other factors potentially related to peak reliabilities on these factors by Spearman’s correlation coefficient (Figure 6). Overall, known peaks contributed to peak reliability, as was expected.

For plasma, all the factors shown in Figure 6 positively influenced the linearity of peak area by dilution, with detection rates being slightly prominent. For urine, such a tendency was not observed.

Restraining within-batch variations contributed to restraining inter-batch variations and vice versa (Figure 6). For known peaks, the variations seemed to be well restrained, but this result largely reflects the known metabolite analysis procedure used in the TMCS; that is, for every batch, area variations of the specific set of known peaks in the QC samples are monitored, and if the variation is too large (beyond mean ± two standard deviations), the corresponding batch is re-analyzed by CE-TOFMS to control the variations. Peak detection rates in the gathered QC samples had a large effect on how variations were restrained, probably because both the peak detection rates and variations reflect how each peak is stably detected. We observed moderate effects of linearity of peak area on restrained variation in plasma but not in urine. The effect on the reliability of whether a peak was clustered with other peaks was not clear.

In summary, using some of these factors as criteria for reliability assessment of peaks, we counted the number of uncharacterized peaks that could be recognized as reliable ones (Table 2). Starting with the 276 and 202 uncharacterized peaks, we obtained 115 and 153 reliable peaks that were clustered into 53 and 114 groups in plasma and urine, respectively. We investigated whether the detected peaks matched known metabolites based on the two procedures described above. First, average migration times and *m/z* values of peak groups in the gathered QC samples (Figure 1b) were compared with those of known metabolites (Table 2, filtering order 2). Second, we checked whether the peaks clustered with known peaks according to the dMTs and the correlation of their peak areas among the samples (Table 2, filtering order 5). In addition, we checked whether the uncharacterized peaks were observed in the standard solution (labeled as “All STD”, which contains 278 and 240 standard compounds for cation and anion, respectively), which is expected to contain most of the available standards (approximately 500 standards), according to migration times and *m/z* values. We found that 18 and 40 peaks with S/N ratios >5 from plasma and urine, respectively, were detected in the standard solution, leaving 35 and 74 peaks that are still uncharacterized (Table 2, filtering order 7).

We used an S/N ratio of >5 as the threshold to extract peaks from the gathered QC samples in the beginning (Table 2, filtering order 1). Among the uncharacterized peaks we finally obtained at the end (Table 2, filtering order 7), 35 (100%) and 69 (93%) of them had S/N ratios >10 in the gathered QC samples, which further confirms their reliability.

## 4. Discussion

We developed a pipeline to assess the reliability of uncharacterized peaks in CE-TOFMS data. Although plasma and urine samples obtained from participants in the Tsuruoka Metabolomic Cohort Study (TMCS) were used in the present study, the developed methods should be applicable to any large-scale metabolomics study that uses clinical samples.

Our pipeline removed a substantial number of unreliable peaks and retained reliable ones. In the first step, peaks were extracted from the gathered QC samples, and a reliability assessment was conducted to estimate the reproducibility of each peak according to the detection rate of the peak in the gathered QC samples. There are several explanations for why some peaks were not reproducible. One of them is noise. Instruments, including those used for liquid chromatography–mass spectrometry (LC–MS) and capillary electrophoresis–mass spectrometry (CE–MS), generate noise in the data from various sources such as random fluctuations in electricity and contaminating chemicals that may be dispersed randomly in the analytes. Noise can generate spurious peaks that may be picked as the real ones or hide authentic peaks that may not be detected. Both cases will reduce the reproducibility of the peaks. Sophisticated algorithms to filter noise from authentic peaks can be useful; however, the most commonly used algorithm relies on the assumption that authentic peaks have Gaussian-like shapes [23,30]. A more empirical approach is need for analyzing CE–MS data, which have peaks with complicated and skewed shapes because of, for example, insufficient sample stacking conditions or fluctuations of electric current during the measurements [29,31]. The incomplete separation of multiple peaks also can impact the reproducibility. For example, ionized isoleucine and leucine have the same *m/z* (132.1019), and their migration times in CE–MS are very similar. Therefore, the two peaks may not be well separated and can be assumed to be a single peak during the peak-picking process. Related to this is variations in retention times (LC) or migration times (CE), especially for CE–MS, where data reliability is greatly affected by variations in migration times for a single metabolite [32]. This will make the accurate alignment (i.e., peak matching) among the samples difficult. Inappropriate alignment may result in incorrect detections of the peaks in some of the samples, thereby reducing the reproducibility.

Next, we assessed whether there was a linear relationship between the amounts of samples and relative peak area, because we expected that this may be useful to distinguish the peaks that originated in actual biological samples from peaks that originated in reagents or from electrical noise. The prominent distributions of correlation coefficients between the amounts of samples and peak areas was located between 0.3 and 1.0 (Figure 3a). We used the value (0.3) as the threshold to filter reliable peaks that may be a quantitative indicator of metabolite concentrations. Although this threshold may be too lenient for small-scale studies, we expect that for large sample sizes, the lenient threshold will pose less of a problem for subsequent statistical analyses using relative peak areas as surrogates for metabolite concentrations.

We also assessed the relationships between the peaks. In particular, correlations of the relative area between pairs of peaks over the samples were calculated, which showed that pairs with high correlation tended to have similar migration times (Figure 5a), which implies that fragmentation of a single metabolite occurred at a specific migration time for each sample. We used this information to cluster peaks according to the inferred fragmentations from a single metabolite and found that areas of peak pairs from plasma showed lower correlations than those from urine. The areas of a peak pair may show a high correlation by factors that include biochemical/biophysical factors and the properties of analytical methods. Because blood flows all around the body, metabolites in blood can be influenced by many different tissues, thus obscuring co-regulation of metabolites in a single tissue. Therefore, we consider that for plasma, biochemical/biophysical factors will be less reflected in peak correlations, and as a result, the effect of metabolite fragmentation will be prominent.

Furthermore, plasma generally contains diverse contaminants (e.g., proteins and lipids) for CE-TOFMS, which potentially could add more noise to the peaks. Additionally, according to the results shown in Table 2, plasma seemed to have more unreliable peaks than urine, and our criteria for removing unreliable peaks were more effective for plasma than they were for urine. We noted that plasma contained a set of peaks with areas that showed inappropriate correlations (correlation coefficient <0) with the corresponding amounts of samples (Figure 3a, plasma).

The efficiency of our proposed criteria and their relationships seem to be complicated (Figure 6). Despite this, we found that, in addition to traditional S/N ratios, the detection rate of each peak in the gathered QC samples may be an efficient criterion for the reliability of the peaks. Linearity between the amounts of samples and peak areas was not always an index of restrained variation and vice versa, which implies that these two criteria are somewhat orthogonal. How to compensate for the variations of uncharacterized peaks (batch corrections) is another problem that needs to be solved.

We focused on the peaks that appeared in the gathered QC samples. Thus, the focus was on peaks that appeared in a broad population. One of our epidemiological interests is to determine how the levels of these metabolites change depending on physiological conditions (i.e., diseases). As an alternative strategy, we can look for peaks that appear specifically in a small subset of a population. However, we speculate that few metabolites are synthesized exclusively in a population subset because, to our knowledge, few reports have shown the existence of such metabolites. This implies that most metabolites are common to a whole population, although their concentrations may greatly differ among individuals depending on their physiological condition.

In the present study, we conducted peak detection from the samples and their alignments automatically without curation. Therefore, the resulting peak areas do not rely on the intuition of curators and should be objective. Additionally, our methods can be applied to metabolomic projects without the need for expert curators. Although the accuracy of the automatic extractions and quantifications of the peaks was a concern, we found a high correlation between the peaks obtained by automatic peak detection and those that were manually curated. Therefore, for peaks with S/N ratios >5 in the gathered QC samples, the accuracy of automatic peak detection and alignment seemed reasonable. We used our proprietary software MasterHands for the automatic peak extraction and annotation, and we are currently improving the software to extract peaks more accurately. For large-scale studies, errors in peak processing may be compensated by the law of large numbers, and better configuration of the instruments may be more important than slightly improving the software performance. In any case, with improvements of the instruments and algorithms, additional peaks with lower intensities that are uncharacterized but reliable may be identified.

The purpose of the current study was to assess reliability of uncharacterized peaks, and we constructed a pipeline for the assessment. There are numerous possible options for the configurations and parameters used in the pipeline, and the selection of software (MasterHands in the current study) or algorithm for the peak extraction is only one of them (Figure 1c). Changing the parameters, such as S/N threshold for peak extraction or distance threshold for clustering peaks, will affect the current results, and we do not consider that the current parameters of our pipeline are optimal for reliable peak extraction. Therefore, the current study is at an exploratory step that will help to optimize the pipeline for future studies. This study also will provide new ideas for the development of other pipelines, especially those in which little attention is paid to the reliable assessment of uncharacterized peaks. The next step in the development of our pipeline will be designing an appropriate evaluation function for reliability assessment of uncharacterized peaks and optimizing its parameters.

Cross-sectional studies of plasma and urine metabolomes have been conducted in the TMCS, and changes in metabolomic profiles associated with alcohol intake [19], metabolite syndrome in postmenopausal females [33], physical activity [34], and diabetes [35] have been reported. However, because the manual curation of peaks is very time consuming, only approximately 115 peaks have been investigated so far. It is likely that many peaks associated with important phenotypes have been overlooked, so expanding the number of peaks, including uncharacterized peaks, that can be investigated may increase the chance of discovering peaks related to important physiological conditions such as disease risks and progressions.

To date, most of the cohort metabolomics studies have used nuclear magnetic resonance, LC–MS, or GC–MS [1]. For example, the Tohoku Medical Megabank Organization (ToMMo), which is geographically close to the TMCS, established a protocol for metabolomic profiling using LC–MS for their cohort study to quantify 270 metabolites [3,36]. The CE-TOFMS that was used in the TMCS has the advantage of identifying ionic metabolites, which are major metabolites in cells. However, the reliability of metabolome data from CE–MS has been of great concern, and recently the data have been investigated from various aspects including migration timing (electrophoretic mobility) [32], and quantification of metabolites at the inter-laboratory [37] and cohort [2] levels. In this study, we described an approach to evaluate the reliability of automatically extracted peaks using the cohort study and proposed a pipeline to filter reliable peaks efficiently.

One of the goals of longitudinal cohort studies using metabolomics is to predict future disease risks based on current metabolomic profiles. The results of this study will contribute to the discovery of uncharacterized peaks that can predict such risks. Identification of metabolites corresponding to uncharacterized peaks is usually very difficult, but the peaks may still be useful for risk prediction even without metabolite identification.

## 5. Conclusions

Our developed pipeline removed substantial numbers of unreliable or redundant peaks and filtered reliable uncharacterized peaks in plasma and urine samples, some of which may correspond to metabolites involved in important physiological processes such as disease progression. We believe that this pilot study gives an idea of the reliability of the peaks when designing large-scale metabolomics studies.

## Figures and Tables

**Figure 1 jcm-10-01826-f001:**
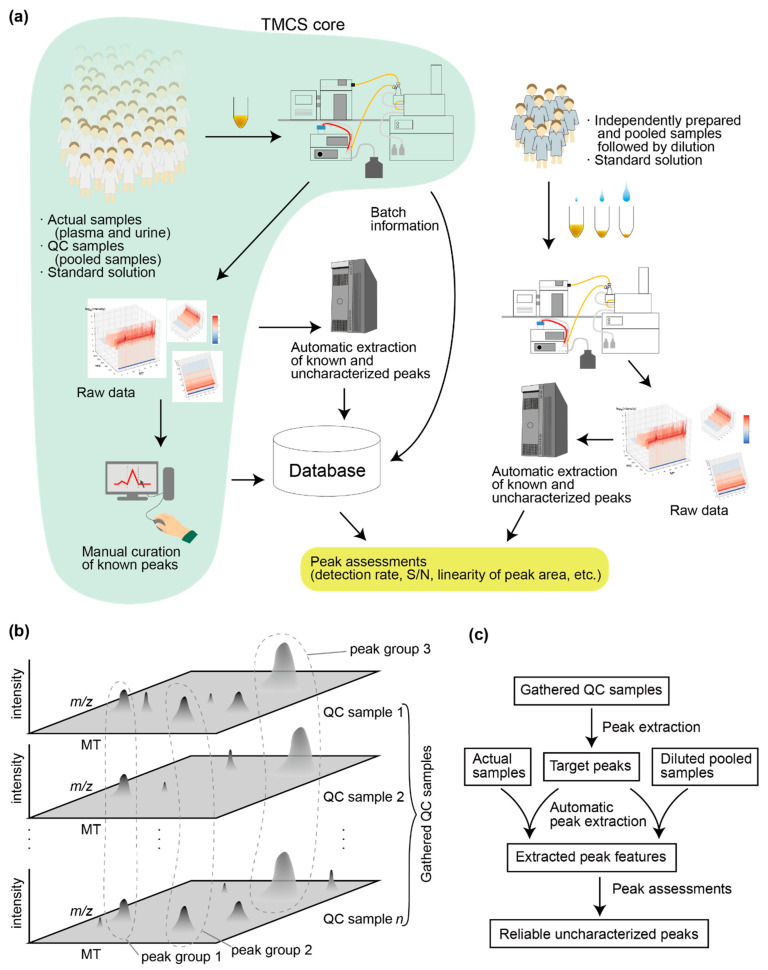
Overview of the pipeline used in the present study and determination of target peaks. (**a**) Overview of the data collection and peak processing/extraction pipeline. The routine used in the Tsuruoka Metabolome Cohort Study (TMCS) is labeled as “TMCS core”. The other procedures are specific to the present study. In the TMCS routine, the characterized peaks are manually curated, whereas in the present study, the peaks were processed automatically. (**b**) Determination of target peaks using 30 randomly selected quality control (QC) samples. The peaks in the 30 gathered QC samples were aligned and clustered to form “peak groups” that had similar *m/z* and migration times (MTs). Ideally, each peak will correspond to a specific metabolite. (**c**) Overview of the peak processing/extraction pipeline focusing on the data flow for filtering reliable uncharacterized peaks. Our proprietary software MasterHands was used for peak extraction.

**Figure 2 jcm-10-01826-f002:**
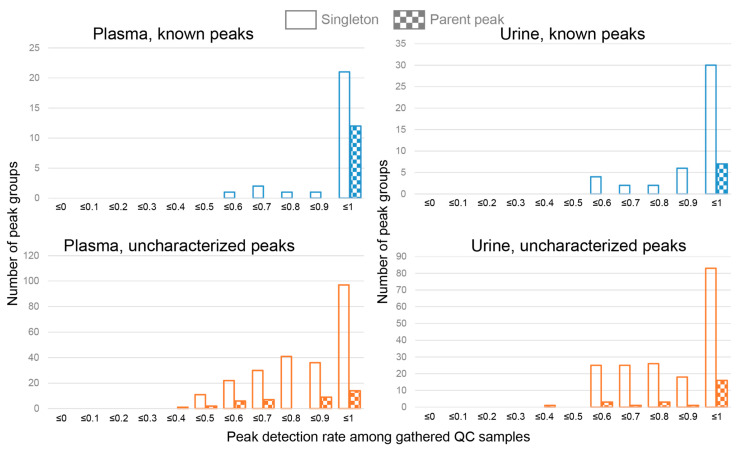
Distribution of peak detection rates among the 30 gathered quality control (QC) samples. Parent peak, a peak that was associated with a potential isotope ion or an adduct ion; singleton, a peak that was not a parent peak.

**Figure 3 jcm-10-01826-f003:**
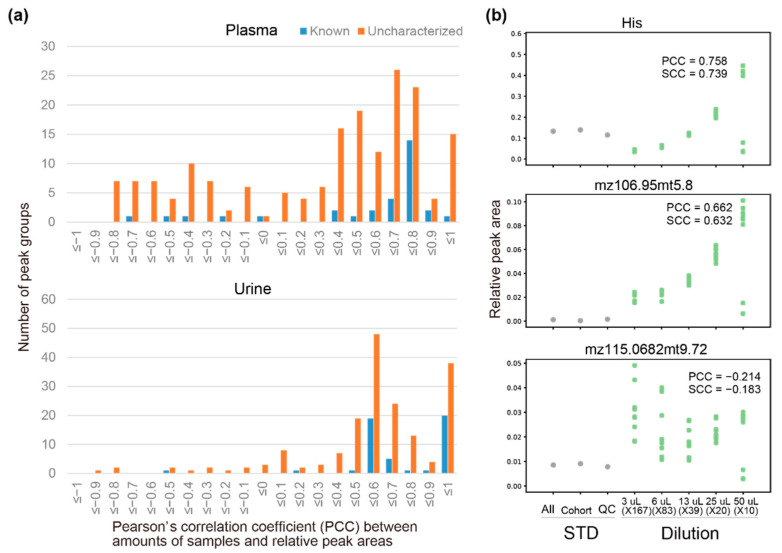
Relationships between the amounts of samples and relative peak areas. (**a**) Distribution of Pearson’s correlation coefficients (PCCs) between the amounts of samples and relative peak areas. Only peaks with signal-to-noise (S/N) ratios >5 in the independently pooled samples were included in this analysis. (**b**) Relationships between the amounts of samples (determined by dilution rates) and the peak areas. PCC and SCC (Spearman’s correlation coefficient) values are shown. Three types (“All”, “Cohort”, “QC”) of standard solutions (STDs) measured together are shown for comparison. “All” contains 278 standard compounds (cations) whereas “Cohort” and “QC”, contain 62 standard compounds (cations). They are also measured in the TMCS routine.

**Figure 4 jcm-10-01826-f004:**
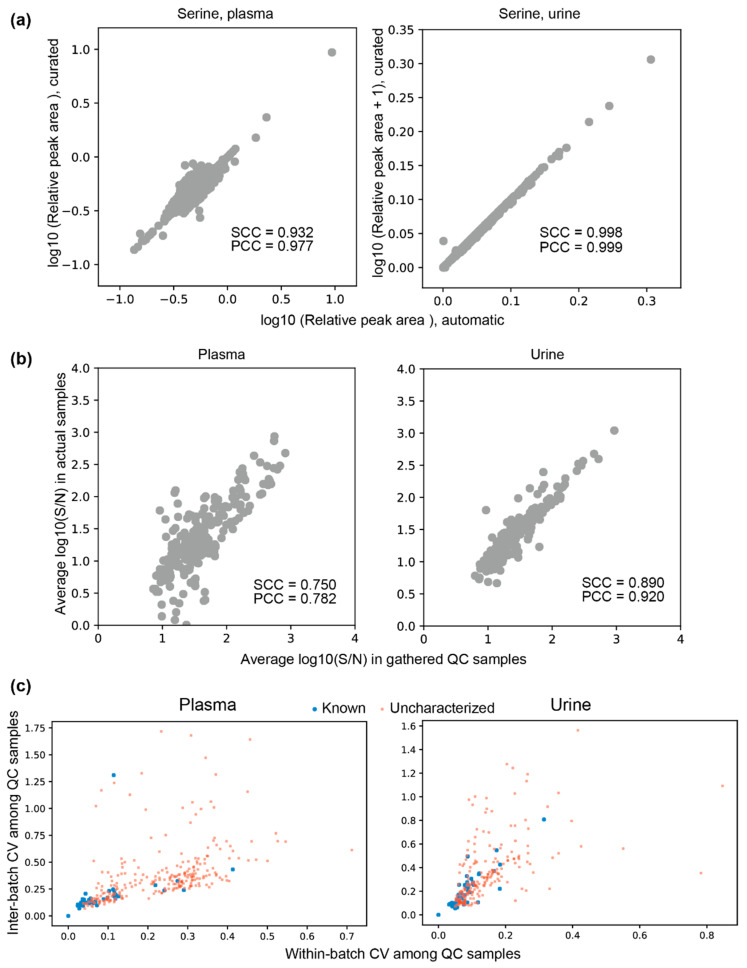
Characteristics of peaks automatically extracted from actual samples and quality control (QC) samples that were measured together within the same batches. (**a**) Plots of relative peak areas that were determined automatically against peak areas that were curated manually. The results for serine are shown. (**b**) Plots of average signal-to-noise (S/N) ratios of uncharacterized peaks in the QC samples against the corresponding S/N ratios in the actual samples. (**c**) Plots of within-batch coefficient of variations (CVs) against inter-batch CVs of relative areas of known and uncharacterized peaks among QC samples. The points at (0,0) indicate the internal standard. SCC, Spearman’s correlation coefficient; PCC, Pearson’s correlation coefficient.

**Figure 5 jcm-10-01826-f005:**
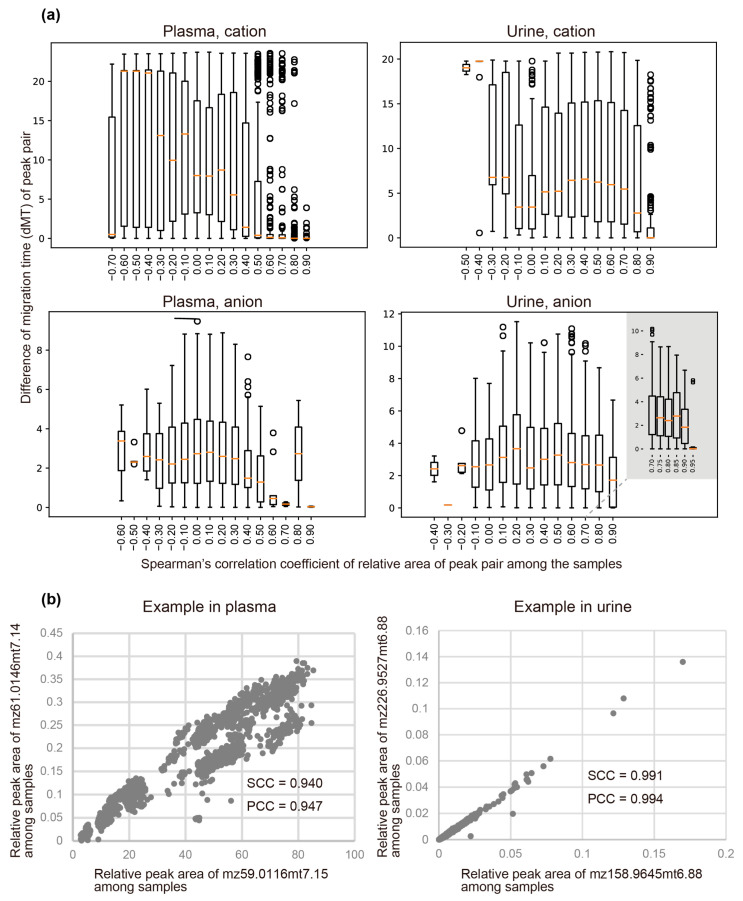
Peak pairs that had close migration times (MTs) whose peak areas showed high correlations over the plasma and urine samples. (**a**) Distribution of differences in migration times (dMT) for peak pairs with a specified range of correlations in their peak areas. (**b**) Representative example of uncharacterized peak pairs that have close migration times and high correlations among samples.

**Figure 6 jcm-10-01826-f006:**
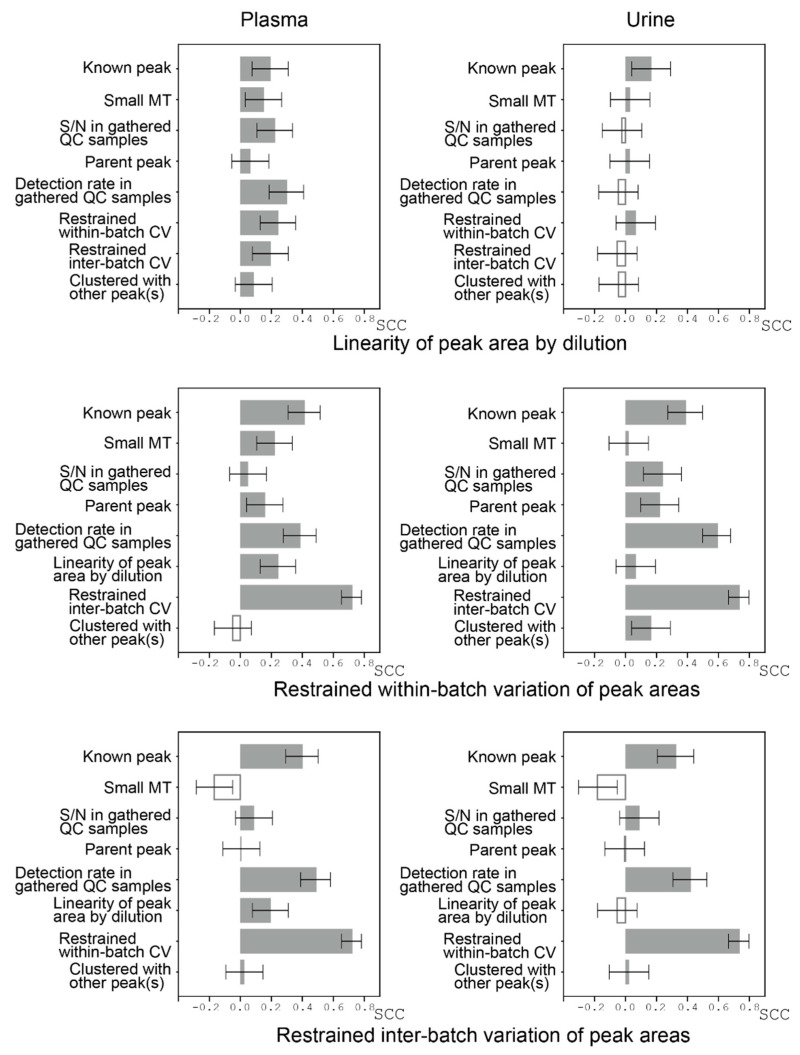
Correlations between factors that may be related to the reliability of the peaks. Spearman’s correlation coefficients (SCCs) between three main reliability factors and other factors were calculated to assess the influence of the other factors on the major factors. The three main reliability factors were linearity of relationship between the amounts of samples (determined by dilution rates) and peak areas (denoted as “Linearity of peak area by dilution”), and how well within-batch and inter-batch variations were restrained. Linearity of peak area by dilution was defined as Pearson’s correlation coefficient between the amounts of samples and the peak areas. The restrained coefficient of variation was defined as −1 × coefficient of variation (CV). The “Known peak” variable was 1 if the peak is known one; otherwise it was 0. “Small MT” was defined as −1 × (migration time). The “Parent peak” variable was 1 if its potential isotope or adduct ion was observed; otherwise it was 0. The “Clustered with other peaks” variable was 1 if the peak was clustered with other peaks; otherwise it was 0. The SCCs are shown as bars with 95% confidence intervals. The filled bars indicate positive SCC values.

**Table 1 jcm-10-01826-t001:** Correlation coefficients between corresponding automatically detected and manually curated peak areas. SCC, Spearman’s correlation coefficient; PCC, Pearson’s correlation coefficient, Tyr, Tyrosine; Pro, Proline; Trp, Tryptophan; Gln, Glutamine; Ser, Serine; His, Histidine; Phe, Phenylalanine; Gly, Glycine; Lys, Lysine; Met, Methionine; Asp, Aspartic acid; Phe-Phe, Phenylalanylphenylalanine. (*) Internal standard.

Plasma	SCC	PCC	Urine	SCC	PCC
o-Acetylcarnitine	0.978	0.978	Carnitine	1.000	1.000
Tyr	0.967	0.973	Proline betaine	0.999	1.000
Pro	0.966	0.976	Lys	0.999	0.999
Trp	0.957	0.963	Gly	0.999	1.000
Choline	0.949	0.951	Guanidinoacetate	0.999	0.999
Ornithine	0.948	0.980	o-Acetylcarnitine	0.999	0.999
Carnitine	0.944	0.953	His	0.999	0.999
Citrulline	0.944	0.959	1-Methylnicotinamide	0.998	0.998
Gln	0.932	0.941	Gln	0.998	0.999
3-Aminopyrrolidine (*)	0.932	0.905	Ser	0.998	0.999
Ser	0.932	0.977	Hypoxanthine	0.997	0.998
Cystine	0.924	0.927	Choline	0.997	0.999
His	0.920	0.963	N6, N6, N6-Trimethyllysine	0.996	0.998
Phe	0.915	0.933	Trp	0.996	0.998
Gly	0.910	0.908	Tyr	0.996	0.998
Lys	0.906	0.917	Guanidinosuccinate	0.995	0.997
Malate	0.867	0.103	Hippurate	0.994	0.981
Met	0.863	0.870	1-Methyladenosine	0.994	0.995
5-Oxoproline	0.849	0.145	Cystine	0.992	0.999
Succinate	0.849	0.108	Homovanillate	0.986	0.983
Guanidinoacetate	0.712	0.267	3-Indoxyl sulfate	0.979	0.972
Taurine	0.585	0.364	Quinate	0.977	0.985
Asp	0.533	0.972	Taurine	0.976	0.986
Proline betaine	0.234	0.634	Pimelate	0.971	0.977
Phe–Phe	0.021	0.121	3-Aminopyrrolidine (*)	0.969	0.989
2-Hydroxypentanoate	−0.099	0.023	2-Hydroxypentanoate	0.963	0.970
Hypoxanthine	−0.194	−0.064	Succinate	0.956	0.971
			N-Acetylaspartate	0.956	0.958
			Uridine	0.939	0.905
			4-Pyridoxate	0.930	0.983
			5-Oxoproline	0.916	0.934
			N-Acetylneuraminate	0.913	0.917
			Isethionate	0.911	0.922
			Threonate	0.896	0.899
			3-Hydroxy-3-methylglutarate	0.856	0.866
			Gluconate	0.772	0.729
			Pro	0.062	−0.008

**Table 2 jcm-10-01826-t002:** Number of reliable uncharacterized peaks detected using seven filtering conditions.

Filtering Order	Filtering Condition	Plasma	Urine
1	Number of peaks with S/N > 5 in the gathered QC samples	340	282
2	Number of peaks whose MTs and *m/z*’s do not match with those of known metabolites	276	202
3	Peaks that are detected in independently prepared QC samples (S/N > 5)	181	180
4	Pearson’s correlation coefficient (PCC) between amounts of samples and relative peak areas is above 0.3	115	153
5	Peaks that are not clustered with known peaks based on the differences in the migration times (dMTs) and the correlation between peak areas among the samples	99	145
6	Inferred fragment clusters based on the differences in the migration times (dMTs) and the correlation between peak areas among the samples	53	114
7	S/N of peak < 5 in the standard solution (labeled as All STD) which contains 278 and 240 standard compounds for cation and anion, respectively	35	74

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
