# Peer review of "Quality Assessment of Untargeted Analytical Data in a Large-Scale Metabolomic Study"

_jcm, 2021, doi:10.3390/jcm10091826_

Round 1

Reviewer 1 Report

In the manuscript, Rintaro et al. set up a new workflow to assess the quality of metabolomic data and tried to sort out uncharacterized peaks in the spectra. The workflow would be useful for future biomarkers identification, as the authors stated. The English of the manuscript reads well.

A few concerns I would like to rise are as follows,

  1. The Introduction is a little lengthy. Some pieces of information were mentioned in the following context, which would be erased.
  2. As we know, if you would like to quantify metabolites from hundreds of samples at the same time, the alignment would be the key factor. Unfortunately, I see nothing about how the authors align the spectra. Such information should be added.
  3. In the material and methods, the authors should state that how they prepared the samples. For MS experiments, sample preparation procedures would be important for the data reproducibility, for example, the final pH of the samples, the dilute ratio of the raw samples and so on. The readers can not rely on the conclusion of the study and the criteria values without such information.

Author Response

We have revised our manuscript “Quality assessment of untargeted analytical data in a large-scale metabolomic study” based on the comments from three reviewers. We responded to each comment raised by the reviewers. Sentences or words in the revised manuscript that are relevant to each response from the authors are marked with comment function of Microsoft WORD. Based on suggestions from reviewer 2, we have done some re-calculations, resulting in broad changes of the results described in figures and texts of the manuscript although they did not affect the overall story of the current study. These are described in the responses to the reviewers. We also improved English in the manuscript by having it proofread by English-editing company (edanz).

Reviewer 1

Comment 1-1: The Introduction is a little lengthy. Some pieces of information were mentioned in the following context, which would be erased.

Response 1-1: We revised the introduction section to make it much shorter (Section Introduction).

Comment 1-2: As we know, if you would like to quantify metabolites from hundreds of samples at the same time, the alignment would be the key factor. Unfortunately, I see nothing about how the authors align the spectra. Such information should be added.

Response 1-2: We added some more details about alignment procedure (Section Materials and Methods - Procedure of automatic peak extraction).

Comment 1-3: In the material and methods, the authors should state how they prepared the samples. For MS experiments, sample preparation procedures would be important for the data reproducibility, for example, the final pH of the samples, the dilute ratio of the raw samples and so on. The readers cannot rely on the conclusion of the study and the criteria values without such information.

Response 1-3: The relevant information has already been published. We briefly wrote its overview and cited the relevant publications (Section Materials and Methods - Data collection and pipeline of computational analyses).

Reviewer 2 Report

Authors made an excellent work trying to reveal the commonly unidentified peaks and whether they are important for drawing conclusions. The method employed by the authors is sound and the overall results are of high importance, both for scientists that carry out metabolomics analyses, but also, for scientists that carry out other type of analysis and deal with big data, as it gives a new way of evaluating their data. 

I have a few, minor comments, that i believe will further improve the manuscript.

  • first of all, the statistical programm used to carry out the statistical analysis should be mentioned. 
  • Why did authors chose the S/N=5 as a criterion and not S/N=10, which is closer to the limits of quantification?
  • Authors calculated both the Spearman’s correlation coefficient and the Pearson’s correlation coefficient.  Why did they calculated both; Is there any reason that the parametric or the non-parametric test should be followed in any specific case; These should be made clear in the text, to avoid confusion. 
  • Error bars should be added in all figures.
  • In Table 1, for most metabolites SCC and PCC values are close. However, there are a few cases, that the two values have a bigger differece. In such cases, which value was selected to draw conclusions?

 After addressing the above issues, i highly recomend the publication of the manuscript.

Author Response

We have revised our manuscript “Quality assessment of untargeted analytical data in a large-scale metabolomic study” based on the comments from three reviewers. We responded to each comment raised by the reviewers. Sentences or words in the revised manuscript that are relevant to each response from the authors are marked with comment function of Microsoft WORD. Based on suggestions from reviewer 2, we have done some re-calculations, resulting in broad changes of the results described in figures and texts of the manuscript although they did not affect the overall story of the current study. These are described in the responses to the reviewers. We also improved English in the manuscript by having it proofread by English-editing company (edanz).

Reviewer 2

Comment 2-1: First of all, the statistical program used to carry out the statistical analysis should be mentioned.

Response 2-1: We added a note about the programs we wrote for this study (Section Materials and Methods - Data collection and pipeline of computational analyses).

Comment 2-2: Why did authors choose the S/N=5 as a criterion and not S/N=10, which is closer to the limits of quantification?

Response 2-2: The threshold of S/N = 5 was determined based on our experience in our institute of extracting uncharacterized peaks that are relatively reliable. Among 35 and 74 reliable uncharacterized peaks which we finally filtered, 35 (100%) and 69 (93%) of them had S/N > 10. Thus changing the threshold to S/N = 10 does not have much effect on our final result. We noted it in the manuscript (Section Results - Peak reliabilities and their relevant factors).

Comment 2-3: Authors calculated both the Spearman’s correlation coefficient and the Pearson’s correlation coefficient.  Why did they calculate both; Is there any reason that the parametric or the non-parametric test should be followed in any specific case; These should be made clear in the text, to avoid confusion.

Response 2-3: Based on the reviewer’s suggestion, we took more care with the usage of Spearman’s correlation coefficient (SCC) and Pearson’s correlation coefficient (PCC). We noticed that the relationship between dilution rates and corresponding peak areas, which we investigated in the initial version of the manuscript was NOT expected to be linear and therefore, PCC was NOT an appropriate metric for assessing correlations. However, we do expect that the relationship between amounts of samples (metabolite concentrations) and corresponding peak areas is linear and PCC would be an appropriate metric in this case. We added this description in Section – Materials and Methods, Dilution of independently prepared and pooled samples. Based on this idea, we did new analyses to investigate this relationship. The results were incorporated into figure 3 and 6. We also made the best effort to show both SCC and PCC for every figure.

Comment 2-4: Error bars should be added in all figures.

Response 2-4: Based on the reviewer’s comment, we added confidence intervals for figure 6. We did not add error bars for figure 2 and 3a since the intention of the figures was NOT to show that the absolute number of peak groups in two classes (singleton vs parent peak for figure 2, and known vs uncharacterized for figure 3a) within each bin was significantly different; Our intention was simply to show the distributions.

Comment 2-5: In Table 1, for most metabolites SCC and PCC values are close. However, there are a few cases, that the two values have a bigger difference. In such cases, which value was selected to draw conclusions?

Response 2-5: The interpretation of the cases where SCC and PCC values are very different is described in “Section Results - Automatic peak extraction from actual samples”.

Reviewer 3 Report

In this manuscript by Rintaro Saito and colleagues, it is difficult to really establish what the authors set out to accomplish. From what I understand, the authors are trying to implement an automated pipeline for the extraction of accurate mass features from a CE-MS data set as their current workflow is limited by the requirement for manual peak extraction. On the whole, it appears that the authors are using an established methodology that is already widely utilised in large-scale metabolomic studies (such as those conducted by 'phenome' centres across the globe). The only novelty that I can ascertain is that the data set is from a CE-MS study but the main analysis is completed by the authors' proprietary software 'MasterHands' which raises the question of how widely used any developments from this manuscript would be. On the whole, I see nothing in this manuscript that advances the field, especially the field of clinically-applied metabolomics and therefore, see no merit in it being published.

Author Response

We have revised our manuscript “Quality assessment of untargeted analytical data in a large-scale metabolomic study” based on the comments from three reviewers. We responded to each comment raised by the reviewers. Sentences or words in the revised manuscript that are relevant to each response from the authors are marked with comment function of Microsoft WORD. Based on suggestions from reviewer 2, we have done some re-calculations, resulting in broad changes of the results described in figures and texts of the manuscript although they did not affect the overall story of the current study. These are described in the responses to the reviewers. We also improved English in the manuscript by having it proofread by English-editing company (edanz).

Reviewer 3

Comment 3-1: From what I understand, the authors are trying to implement an automated pipeline for the extraction of accurate mass features from a CE-MS data set as their current workflow is limited by the requirement for manual peak extraction.

Response 3-1: We are trying to implement an automated pipeline for the extraction of reliable uncharacterized peaks and the extraction of accurate features (i.e., migration time, m/z, annotation, etc) from the raw data that the reviewer mentioned is only one piece of our pipeline. We clarified this point in the manuscript based on the reviewer’s comment (Section Discussion).

Comment 3-2: On the whole, it appears that the authors are using an established methodology that is already widely utilized in large-scale metabolomic studies (such as those conducted by 'phenome' centres across the globe). The only novelty that I can ascertain is that the data set is from a CE-MS study but the main analysis is completed by the authors' proprietary software 'MasterHands' which raises the question of how widely used any developments from this manuscript would be.

Response 3-2: For the peak extraction from raw data, we are using an already established method as pointed out by the reviewer, although it is just one piece of our pipeline for assessing reliability of uncharacterized peaks. Also we are not sure whether our pipeline would be widely used, as the reviewer mentioned, since we are not sure whether our current pipeline would be the best one for reliable peak extraction. These two points were briefly mentioned in the text as the limitations in our study (Section Discussion). We still believe that the present study will help readers to get some idea of how to develop their own pipeline since little attention has been paid on reliability assessment of uncharacterized peaks.

Round 2

Reviewer 3 Report

Although the authors have made changes to their manuscript, these appear to be cosmetic in nature and have not addressed my original concerns. The exact purpose and focus of the manuscript remains unclear and appears to add minimal, if any, scientific value to the community - something which the authors appear to agree with based on their response.

Author Response

Comment 3-1: From what I understand, the authors are trying to implement an automated pipeline for the extraction of accurate mass features from a CE-MS data set as their current workflow is limited by the requirement for manual peak extraction.

Response 3-1: According to the reviewer’s comment, we realized that the purpose of our study described in the text was confusing. Accordingly, we made a new figure (Figure 1c) which clarifies our purpose and described it in the introduction. We are trying to implement an automated pipeline for the extraction of reliable uncharacterized peaks (Figure 1c), and the extraction of accurate features (i.e., migration time, m/z, annotation, etc) from the raw data (which corresponds to “Automatic peak extraction” in Figure 1c) that the reviewer mentioned is only one piece of our pipeline.

We also noticed that the following statement in the introduction in the previous version of the manuscript was confusing if it is taken as our main purpose:

“In the present study, we developed a pipeline to automatically extract peaks from given samples …”

Accordingly, we changed the above to:

In the present study, we developed a pipeline to assess reliabilities of automatically extracted peaks from given samples (Figure 1).

Comment 3-2: On the whole, it appears that the authors are using an established methodology that is already widely utilized in large-scale metabolomic studies (such as those conducted by 'phenome' centres across the globe). The only novelty that I can ascertain is that the data set is from a CE-MS study but the main analysis is completed by the authors' proprietary software 'MasterHands' which raises the question of how widely used any developments from this manuscript would be.

Response 3-2: We agree with the reviewer that our current work raises the question of how widely our pipeline would be used. Accordingly, we mentioned that the development of our pipeline is at an exploratory step in the discussion. We also described that the peak extraction software is just one piece of our pipeline for assessing reliability of uncharacterized peaks in the discussion and newly created Figure 1c. We note that MasterHands is widely used software for processing peaks from CE-MS. In fact, Google scholar outputs ~244 articles with keywords “ce-ms”, “metabolomics”, and “masterhands” (As of April 12, 2021).

Statements or words in the manuscript that are relevant to each response from the authors are marked as “2nd revision, response 3-X” with comment function of Microsoft WORD.